# Whole-exome sequencing of epithelial ovarian carcinomas differing in resistance to platinum therapy

Viktor Hlaváč[1,2] , Petr Holý[1,2,3] , Radka Václavíková[1,2], Lukáš Rob[4] , Martin Hruda[4], Marcela Mrhalová[5], Petr Černaj[6], Jiří Bouda[6], Pavel Souček[1,2]

**Epithelial ovarian carcinoma (EOC) is highly fatal because of the risk of resistance to therapy and recurrence. We performed whole-exome sequencing of blood and tumor tissue pairs of 50 patients with surgically resected EOC. Compared with sensitive patients, platinum-resistant patients had a significantly higher somatic mutational rate in *TP53* and lower in several genes from the Hippo pathway. We confirmed the pivotal role of somatic mutations in homologous recombination repair genes in platinum sensitivity and favorable prognosis of EOC patients. Implementing the germline homologous recombination repair profile significantly improved the prediction. In addition, distinct mutational signatures, for example, SBS6, and overall mutational load, somatic mutations in *PABPC1*, *PABPC3*, and *TFAM* co-segregated with the resistance status, high-grade serous carcinoma subtype, or overall survival of patients. We generated germline and somatic genetic landscapes of prognostically different subgroups of EOC patients for further follow-up studies focused on utilizing the observed associations in precision oncology.**

## Introduction

Ovarian cancer (OMIM: 167000) is the second most frequent cause of death among gynecological cancers. GLOBOCAN cancer statistics for the year 2020 show that ovarian cancer is the eighth most common (313,959 new cases) and fatal (207,252 deaths) cancer diagnosis among women worldwide (Sung et al, 2021). It is usually diagnosed at an advanced stage (FIGO III or IV) when the 5-yr survival rate is ~20–45%, whereas it is 40–70% for stages I or II (Matz et al, 2017; Ovarian Cancer Survival Rates, 2020 at www.cancer.org). The most frequently diagnosed type is epithelial ovarian carcinoma (EOC) (90%). EOC comprises type I (endometrioid, mucinous, clear cell, and low-grade serous ovarian carcinomas [LGSCs]) and type II (high-grade serous ovarian carcinomas [HGSCs], carcinosarcomas, and undifferentiated carcinomas). Type II carcinomas account for 70% of all cases (Matz et al, 2017; Ovarian Cancer Survival Rates, 2020 at www.cancer.org).

Most of the patients undergo chemotherapy combining platinum derivatives (carboplatin or cisplatin) and taxanes (paclitaxel or docetaxel) (Kim et al, 2012). The recently introduced targeted therapies, for example, poly(ADP-ribose) polymerase (PARP) inhibitors, such as olaparib, or antiangiogenic agents such as bevacizumab or pazopanib seem to improve the prognosis of the patients (Cortez et al, 2018; Lisio et al, 2019). However, a late diagnosis at advanced stages of the disease increases the chance of chemotherapy resistance and the development of metastases. Thanks to the employment of new robust techniques, for example, next-generation sequencing, researchers can characterize the roles of DNA, RNA-coding elements, and noncoding elements in the development and progression of cancer (Seborova et al, 2021). Genomics research has brought the first useful biomarkers for the management of EOC patients, that is, patients bearing *BRCA1*/*BRCA2* mutations being sensitive to treatment with PARPi (Chartron et al, 2019). Other promising biomarkers and tailored therapies exploiting for example defects in the DNA repair system of tumor cells are on the way (Pilié et al, 2019).

Recently, EOC tumors were classified also by genetic variability—into two major subtypes. Rojas et al (2016) reported somatic gene mutation profiles for type I EOC (mutations in the MAPK pathway—*KRAS*, *BRAF*, *PTEN*, *CTNNB1*, etc.) and type II EOC (*TP53*, *BRCA1*, *BRCA2*, *KIT*, and *EGFR*) (Rojas et al, 2016). Clinical and molecular characteristics of type I and II EOC may become novel predictive or prognostic biomarkers and eventually therapeutic targets (Kurman & Shih, 2016). As recently surveyed, DNA repair machinery also plays an important role in the EOC development risk, prognosis, and therapeutic outcomes (Tomasova et al, 2020).

Previous whole-exome sequencing (WES) studies focused on either hereditary breast and ovarian cancer (Felicio et al, 2021) or

[1]Biomedical Center, Faculty of Medicine in Pilsen, Charles University, Pilsen, Czech Republic   [2]Toxicogenomics Unit, National Institute of Public Health, Prague, Czech Republic   [3]Third Faculty of Medicine, Charles University, Prague, Czech Republic   [4]Department of Gynecology and Obstetrics, Third Faculty of Medicine, Charles University and University Hospital Královské Vinohrady, Prague, Czech Republic   [5]Department of Pathology and Molecular Medicine, Second Faculty of Medicine and Motol University Hospital, Charles University, Prague, Czech Republic   [6]Department of Gynecology and Obstetrics, Faculty of Medicine and University Hospital in Pilsen, Charles University, Pilsen, Czech Republic

Correspondence: pavel.soucek@szu.cz

somatic alterations (Li et al, 2019; Sztupinszki et al, 2021). Pathogenic germline variants were found in *CHEK2*, *MUTYH*, *PMS2*, and *RAD51C*, whereas rare pathogenic variants were observed in DNA repair and other cancer-related genes in a Brazilian study (Felicio et al, 2021). Another study using The Cancer Genome Atlas (TCGA) database assessed mutational signatures induced by homologous recombination deficiency in EOC patients. Interestingly, quantification of large chromosomal alterations was equally as efficient in identifying homologous recombination deficiency in EOC as methods incorporating genomic aberrations (Sztupinszki et al, 2021). Finally, chemotherapy-resistant EOC tumors showed higher prevalence of *MYC* and *PIK3CA* amplification, and these changes persisted during progression to metastasis and were even more significant in tumor recurrence (Li et al, 2019). Despite the above, information about the genetic signature of resistance to therapy and prognosis of EOC patients, which could be used for targeted treatment to limit further cancer spread, is still very scarce.

The present study aimed to analyze whole-exome somatic mutational spectra of the prospectively collected cohort of surgically resected patients, treated with platinum derivative–based regimens. We compared genetic alterations in individual genes and pathways, copy number variations (CNVs), and mutational signatures in patients stratified by the platinum resistance status, tumor subtype, and overall survival (OS), providing a lead for follow-up studies focused on utilizing this somatic profile in precision oncology of EOC.

# Results

## Clinical characteristics of the patients

The main clinical characteristics of all patients are in Table 1. The median age of patients at diagnosis was 60 years (range 40–81); about 90% of patients had advanced disease with FIGO stage III or IV, and most of the patients (70%) had the HGSC subtype and histological grade 3 (78%). Four patients presented with distant metastases, and the overall frequency of residual disease after surgery was 56%. Thirty patients had PFI ≥ 12 mo and were classified as platinum-sensitive, whereas 20 patients experienced recurrence or progression of the disease within 12 mo after completion of platinum-based chemotherapy (resistant group). Four of the platinum-resistant patients had PFI 7–10 mo, and one had FIGO stage IV, whereas 15 had tumor residuum after surgery (75%). Besides a higher proportion of residuum after surgery among resistant patients ($P$ = 0.027), the compared groups of patients did not significantly differ in other clinical characteristics (Table S1). The median follow-up time of the patients was 34.5 ± 29.5 mo, and 35 patients (70%) died during this follow-up. The median OS of resistant patients was highly significantly shorter (16 mo) than that of sensitive patients (68 mo) ($P$ = 2.1 × 10$^{-13}$).

## General descriptors of the WES and variant profiling

The average coverage of sequenced regions with distinct reads in tumor samples was 243 ± 29 (median 245). 97.7% of bases were

**Table 1. Clinical characteristics of the patients.**

| Parameters | Number of patients | % |
|---|---|---|
| Age at diagnosis, median±SD (years) | 60.0 ± 9.7 | 100 |
| FIGO stage | | |
| I | 2 | 4 |
| II | 3 | 6 |
| III | 40 | 82 |
| IV | 4 | 8 |
| Not available | 1 | — |
| Histological grade (G) | | |
| G1 | 5 | 10 |
| G2 | 6 | 12 |
| G3 | 39 | 78 |
| Tumor subtype | | |
| HGSC | 35 | 72 |
| Mucinous | 5 | 10 |
| LGSC | 4 | 8 |
| Clear cell | 3 | 6 |
| Endometrioid | 2 | 4 |
| Not available | 1 | — |
| Distant metastasis | | |
| Absent | 45 | 92 |
| Present | 4 | 8 |
| Not available | 1 | — |
| Residuum after surgery | | |
| Absent | 22 | 44 |
| Present | 28 | 56 |
| Resistance status | | |
| Sensitive | 30 | 60 |
| Resistant | 20 | 40 |

FIGO, Fédération Internationale de Gynécologie et d'Obstétrique; HGSC, high-grade serous carcinoma; LGSC, low-grade serous carcinoma; SD, standard deviation.

covered at least 30×, and 95.6% of bases were covered at least 50×. The average coverage of sequenced regions in blood samples was 35 ± 6 (median 34), and 91.5% of bases were covered at least 10×.

The total number of somatic variants per sample was 1,781 ± 282 on average (ranging from 1,161 to 2,386, median 1,754). To minimize the number of false-positive calls, somatic variants were then filtered based on selected quality filtration criteria (see the Materials and Methods section: Raw data preprocessing and variant detection). On average, 478 ± 133 variants per sample passed this filtering (251–818, 455). We additionally applied functional prediction by the SnpEff tool and received on average 28 ± 12 variants with highly probable functional effect (11–63, 27) and 140 ± 52 variants with moderate predicted impact (55–265, 136) per patient. Counts of somatic variants for all patients are in Table S2. The most common class of variants was a missense mutation (Fig 1A) and the most

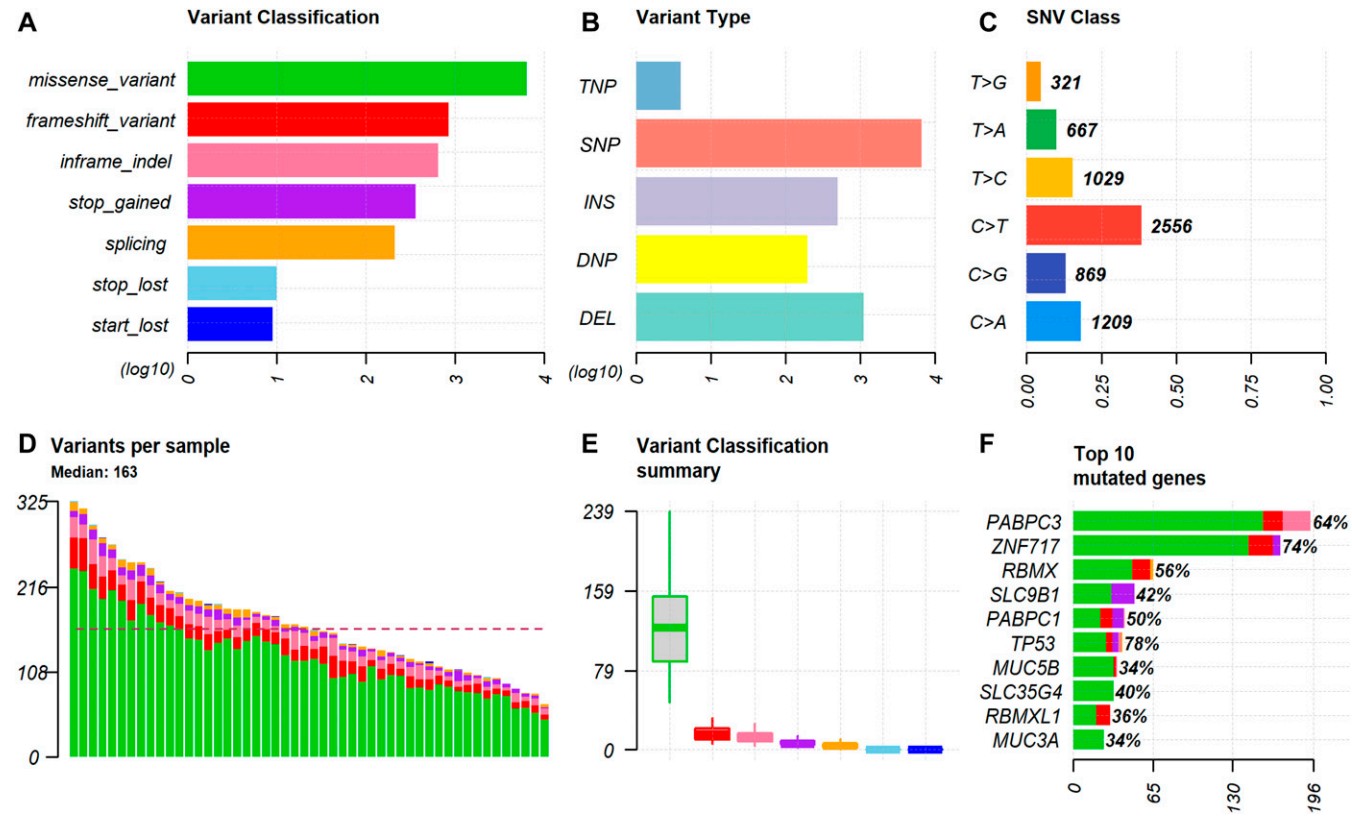

**Figure 1.   Summary of the distribution of somatic variants in all epithelial ovarian carcinoma patients.**
Footnotes: This figure shows the overall distribution of the variants. Only variants with predicted moderate or high protein impact are shown. **(A)** The classification of variants according to the functional effect with counts on the x-axis. **(B)** The type of the variants (TNV stands for trinucleotide variant; SNP, single-nucleotide variant; INS, insertion; DNV, dinucleotide variant; DEL, deletion). Both (A) and (B) have x-axis in the $\log_{10}$ scale to visualize also less frequent variants. **(C)** The type of nucleotide change. **(D)** The counts and distribution of the variants for the indicated samples; dashed line represents the median (163 variants per sample). **(E)** The box and whisker plots of the variant classes. Color coding corresponds to the classification of variants in top left. **(F)** Top 10 genes with the highest numbers of variants are shown on the x-axis, and percentages of patients harboring any variant in the indicated genes are shown next to the bars.

frequent type was a single-nucleotide variant (SNV, Fig 1B), followed by deletion and insertion. The most common nucleotide change was the C to T transition (Fig 1C). The top mutated genes were *TP53*, *ZNF717*, *PABPC3*, and *RBMX* with somatic mutational events in more than 50% of patients (Fig 1F).

Next, we filtered out somatic variants in genes listed in the FrequentLy mutAted GeneS (FLAGS) database (Shyr et al, 2014). The top 20 genes scored by FLAGS were excluded from further analyses (Table S3). Oncoplot of the list of genes with the highest counts of variants in all EOC patients is in Fig S1. The most frequently mutated genes in the complete EOC patient set remained *TP53*, *ZNF717*, *PABPC3*, and *RBMX* (all above 50%).

Finally, single-base substitution (SBS) mutational profiles (portraits) were constructed for all tumor samples (Fig S2). Among the identified 78 SBSs, 25 signatures had <10% of nonzero values across all tumor samples (SBS4, SBS5, SBS7b, SBS9, SBS10a, SBS10c, SBS11, SBS14, SBS26, SBS28, SBS31, SBS35, SBS36, SBS38, SBS40, SBS41, SBS45, SBS47, SBS48, SBS55, SBS56, SBS60, SBS85, SBS92, and SBS93) and were excluded. The rest of the SBSs (n = 53) were considered the most informative for comparing their distribution among patients divided by main clinical data. The distribution of all SBSs across the patient set is in Table S4. For survival analyses, we

further divided patients into groups with zero versus nonzero shares of the abovementioned signatures.

CNVs were also called using paired tumor-matched samples. For this, we used read depth and B-allele frequency methods implemented in the ExomeDepth tool. As the theoretical level of tumor sample purity/clonal fraction for signal thresholding, we used the value of 40% (see the Materials and Methods section: Raw data preprocessing and variant detection). CNVs with lower clonal fractions therefore may not have been discovered. However, based on the distributions of somatic variant allele fraction in analyzed samples, we assumed the sample purity to be equal to or above 40% (data not shown). Most of the clonal CNVs therefore should have been effectively detected. The average count and size of CNVs per tumor sample were 159 ± 99 (24–419, median 136) and 8,170,715 ± 4,179,088 (745,937–21,101,100, median 7,650,205), respectively. When divided by the number of copies, on average, three CNVs with zero copies or 50 (median 46) with a single copy (deletions) were observed per patient. Analogously, 78 (66) gains with three copies and 29 (22) with four or more copies were detected per patient on average (Table S5).

Besides the variation in tumor samples, we also detected short germline variants in the blood samples of the patients. Similarly to

somatic variants, we performed variant calling and filtering in agreement with GATK best practices (see the Materials and Methods section: Raw data preprocessing and variant detection for more details). On average, 39,054 ± 793 germline variants per patient were identified (36,523–40,282, median 39,132). After filtering, 33,992 ± 1,495 variants per patient (28,739–36,238, median 34,052) remained. Functional prediction using SnpEff resulted in 253 ± 15 variants with highly probable functional effect (224–287, median 254) and 8,804 ± 273 variants with moderate predicted impact (7,793–9,212, median 8,841) per patient. Germline variants are in Table S6. Because of the limited study size, we did not analyze germline variants further, except for the analysis of the homologous recombination repair (HRR) genes, presented in a later section of the Results.

### Genetic profiles of patients divided by the platinum resistance status

We divided patients into subgroups according to their platinum resistance status and compared their somatic mutational profiles (Fig 2).

In the platinum-resistant subgroup of patients, the frequency of mutations was significantly higher for *TP53* (*P* = 0.033) and lower for *PABPC3* (*P* = 0.035). None of these associations remained significant after the false-discovery-rate (FDR) adjustment. Although notably higher in sensitive patients, the frequency of somatic alterations in *FRG2B* was not significantly different (*P* = 0.067). Similarly, among the less frequently altered genes, *KDM6B*, *MAGEL2*, *PI4KA*, and *SLIT1* alterations were found only in the sensitive subgroup of patients (all *P* = 0.075; Table S7).

Apart from the frequency, the spectrum of *TP53* mutations also considerably differed between both subgroups of patients. The platinum-resistant patients had a much higher proportion of functionally important types of alterations (high impact, i.e., stop-gained or frameshift) than the sensitive subgroup (35% versus 13%; Fig 3). In addition, the number of alterations with moderate impact (including missense and inframe indels) was notably higher (95% versus 67%; Fig S3).

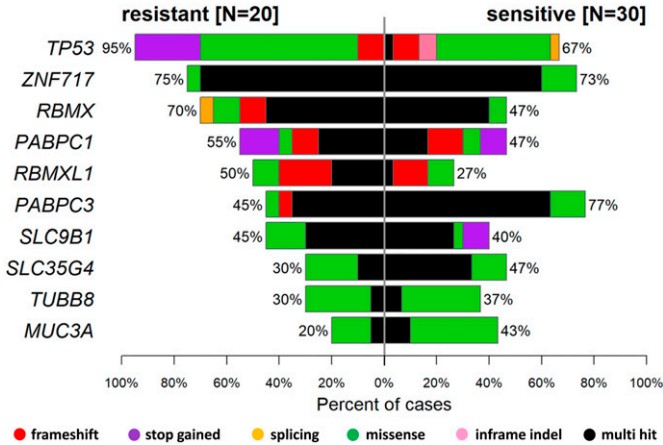

**Figure 2. Co-bar plot of 10 genes with highest counts of variants with moderate or high impact in epithelial ovarian carcinoma patients divided by the platinum resistance status.**

Analysis of co-occurrence (genes mutated together in the same sample) revealed that in the platinum-resistant subgroup, mutations in *PABPC3* significantly co-segregated with *PABPC1* alterations, whereas no significant mutual exclusivity (the opposite of co-occurrence) was observed. On the other hand, in the sensitive patients, mutations in *TP53* were highly significantly co-segregated with the FLAGS *TTN* and less significantly with another FLAGS *MUC5B*. No mutual exclusivity was again apparent (Table S8 and Fig S4).

Furthermore, we performed pathway analysis using several tools. After comparing mutation rates in 10 established cancer pathways (Sanchez-Vega et al, 2018, Fig S5), we found *TP53* (*P* = 0.037), apparently because of the differential mutational profile of *TP53* alone, and Hippo (*P* = 0.042; Table S9 and Fig S5) to be differentially mutated. Patients with somatic *TP53* mutations had an odds ratio (OR) of 7.87 toward being platinum-resistant (95% confidence interval 0.94–374.58). In contrast, patients with mutations in the Hippo pathway had an OR of 0.26 for platinum resistance (95% CI = 0.06–0.96; Fig 4). Panther analysis of GO terms (molecular function, biological process, and cellular component) failed to reveal notable differences between patients divided by the platinum resistance status (Fig S6). KEGG pathway exome-wide analysis using unique sets of mutated genes (Table S10) also did not suggest differentially altered pathways or diseases specific for the platinum-resistant subgroup, besides perhaps thyroid carcinoma (Table S11). Genes connected with the terms Huntington's disease, Parkinson's disease, Alzheimer's disease, prion disease, spinocerebellar ataxia, spermatogenic failure, early infantile epileptic encephalopathy, hereditary spastic paraplegia, X-linked mental retardation, autosomal recessive mental retardation, and Charcot–Marie–Tooth disease seemed overrepresented in sensitive patients (Table S11).

The comparison of tumor mutation burden (TMB) (n = 50) with the ovarian carcinoma TCGA-OV dataset (n = 411) showed a slightly higher TMB in our dataset. Between subgroups of our cohort defined by resistance status, there was no notable difference in TMB (Fig S7). We also performed cancer driver prediction using MutSigCV and TCGA OV-TP dataset for comparison. This analysis revealed an 83% *TP53* mutational rate in the TCGA dataset compared with 78% (including silent mutations) in our dataset. The rest of the genes had lower mutational rates in the TCGA than in our dataset (≤5%; Fig S8).

Except for SBS6, the distribution of the evaluated mutational signatures did not significantly differ between resistant and sensitive patients (*P* > 0.05; Table S12). Resistant patients had significantly higher SBS6 share than sensitive patients (*P* = 0.019), although this association did not pass the FDR adjustment.

The average count and size of CNVs per tumor sample did not significantly associate with the resistance status of patients (*P* > 0.05; Table S12).

### Genetic profiles of patients divided by other clinical factors

We further analyzed somatic mutational profiles, CNVs, and mutational signatures in patients stratified by major clinical factors (FIGO stage, histological grade, tumor subtype, the presence of distant metastasis, and residuum after surgery). Of these clinical

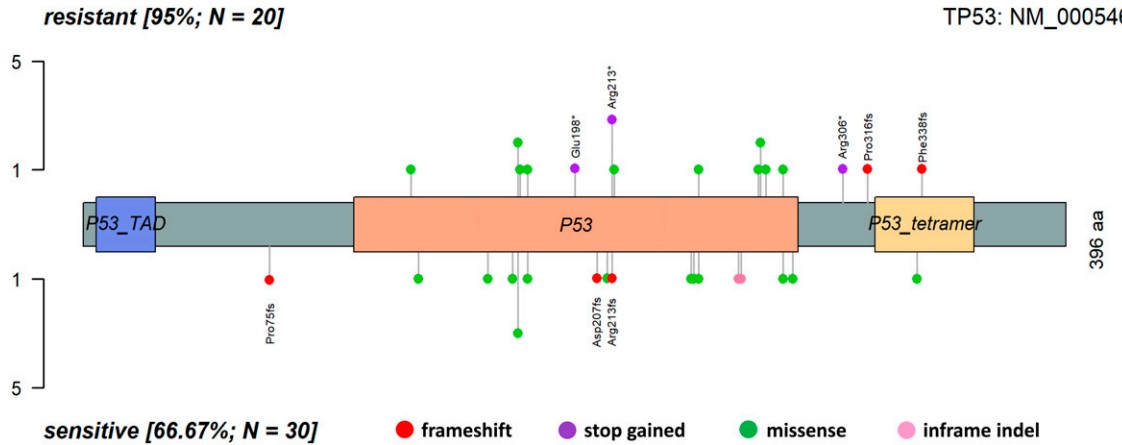

**Figure 3. Lollipop plot of topology of *TP53* alterations with moderate or high impact in platinum-resistant (upper part) and -sensitive (bottom part) epithelial ovarian carcinoma patients.**

factors, only stratification of patients into groups with the HGSC subtype (n = 35) compared with patients with other subtypes (LGSC, mucinous, clear cell, and endometrioid, n = 14; one sample with an undefined subtype excluded) revealed significant associations. The number of all somatic variants and those passing all filters per sample was significantly higher in tumors of the HGSC subtype than in tumors with other subtypes (mean ± SD, 1,837 ± 269 versus 1,645 ± 289, $P = 0.019$ and 310 ± 91 versus 228 ± 130, $P = 0.007$, respectively; Table S12).

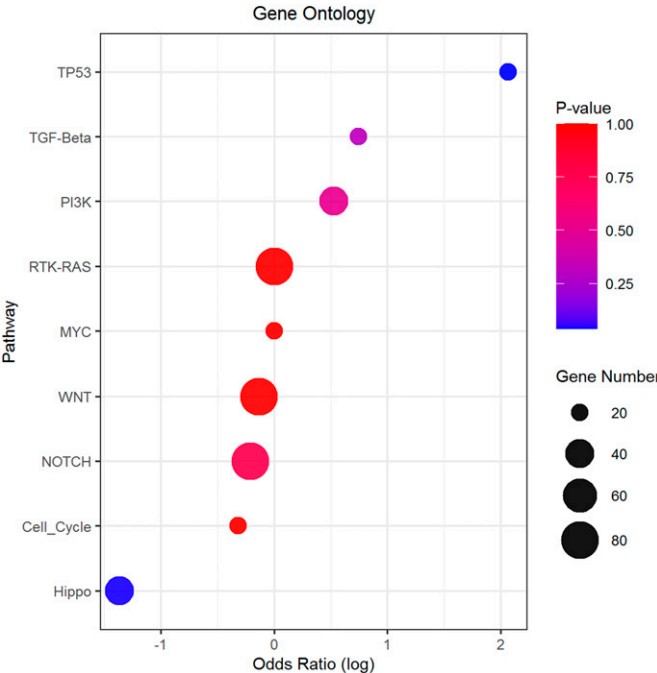

**Figure 4. Enrichment analysis of top 10 canonical oncogenic pathways in patients divided by platinum resistance status.**
Footnotes: Odds ratios of resistant versus sensitive patients by the Fisher test are on x-axis (log transformed) and pathway names on the y-axis. Gene number: total number of genes in each pathway.

A plot of mutated genes with the highest counts of variants in EOC patients divided by the EOC tumor subtype is in Fig 5.

The HGSC subtype had rather lower somatic mutation rates in *PABPC1*, *ZNF717*, *RBMX*, *SLC9B1*, or *TFAM*. Patients with somatic *TP53* mutations showed an OR toward having the HGSC subtype 7.34 (95% CI = 1.42–44.93, $P = 0.007$). However, this association did not withstand the FDR adjustment (Table S13). On the other hand, the association on the *TP53* pathway level gained OR 9.98 (95% CI = 1.76–75.39) toward the HGSC subtype and was significant in both crude and FDR adjusted analyses ($P = 0.003$ and $P = 0.030$, respectively; Table S14).

Patients with the HGSC tumor subtype had a higher share of SBS34 ($P = 0.046$) and a lower share of SBS17b ($P = 0.049$) than patients with other EOC subtypes, and metastatic patients had a significantly higher share of SBS88 than the rest of patients ($P = 0.016$) (Table S12). None of these associations remained significant after the FDR adjustment.

The HGSC subtype also had a higher number and length of somatic CNVs than the rest (number – 179 ± 83 versus 113 ± 126 and length – 8,903,306 ± 3,838,706 versus 5,938,660 ± 4,249,685) ($P = 0.004$ and $P = 0.024$, respectively). Moreover, HGSC subtypes had significantly more single-copy deletions ($P = 0.006$) and more gains in general (regardless of the copy number; $P = 0.007$ for three and $P = 0.028$ for four or more copies) than the rest of the patients. CNVs were not associated with the rest of the clinical factors (all results are in Table S12).

## Survival analysis

Patients with a higher than the median number of somatic variants passing all filters had significantly longer OS than the rest ($P = 0.021$; Fig 6A). The average number and length of somatic CNVs were not associated with OS ($P > 0.05$), but patients with a lower number of gains (three or four and more copies) than the median (66 and 22, respectively) had significantly poorer OS than the rest ($P = 0.016$ and $P = 0.038$, respectively; Fig 6B and C). In addition, patients with nonzero values for SBS6 had significantly poorer OS than patients with zero values ($P = 0.002$; Fig 6D). A trend toward poorer OS was

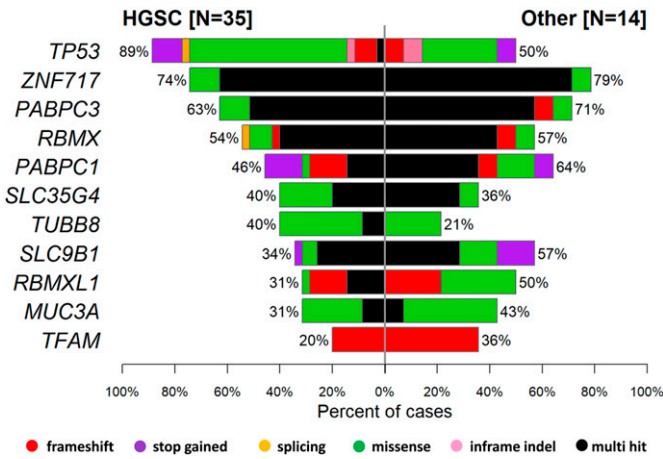

**Figure 5. Co-bar plot of mutated genes with highest counts of variants with moderate or high impact in patients divided by the epithelial ovarian carcinoma tumor subtype.**

also observed in patients with nonzero values for SBS10b and SBS21 (*P* = 0.050, Fig S9, and *P* = 0.034, Fig S9, respectively).

### Importance of mutations in genes connected with HRR

In the end, we stratified patients by the occurrence of germline or somatic alterations whose functional importance was classified as HIGH, in 16 HRR genes to subgroups as defined elsewhere (Norquist et al, 2018). We distinguished between *BRCA1/2*-only–mutated and non-BRCA gene–mutated patients (Table S15). Patients with somatic or germline mutations in HRR genes evaluated separately were significantly more frequently platinum-sensitive than patients without these mutations, and this association was even more robust when these alterations were grouped (Table 2).

Furthermore, patients with somatic alterations in any of the 16 HRR genes had significantly longer OS than the rest of the patients (*P* = 0.047; Fig S10), whereas this effect was nonsignificant in patients divided by germline alterations in HRR genes (*P* = 0.152). Patients with somatic or germline alterations in *BRCA1/2* genes also had significantly longer OS than the rest of the patients (*P* = 0.030; Fig S10), whereas this was not true for patients with non-BRCA alterations (n = 6, *P* > 0.05). Most importantly, patients with somatic or germline alterations in HRR genes had highly significantly longer OS (*P* = 0.006; Fig 7).

## Discussion

EOC is a deadly disease, and prognostic or predictive biomarkers, allowing truly personalized therapy for patients, are very slowly coming into the clinical decision-making process. We aimed to generate a pilot dataset and provide a comprehensive comparison of genetic alterations within the coding part of the genome of EOC patients stratified by their platinum resistance status, tumor subtype, and OS.

Our data analysis shows that patients resistant to chemotherapy based on platinum and taxane combination have a significantly

higher load of somatic mutations in the *TP53* gene than sensitive patients. This association was demonstrated on both single-gene and whole-pathway levels, although the association on the pathway level can be attributed almost entirely to alterations in the *TP53*. As a result, patients with *TP53* mutations had an almost eight times higher odds ratio toward treatment resistance. Moreover, upon closer inspection of predicted functional effects of these mutations, resistant patients had mutations more frequently classified as "high impact," that is, mainly nonsense or frameshift loss of function alterations, compared with sensitive patients.

*TP53* is an important tumor suppressor gene coding the nuclear p53 protein essential for DNA damage sensing; namely, it is responsible for the initiation of the phosphorylation cascade, leading to the cell cycle stall in the G1-S phase transition. Functional p53 helps the cell get enough time to repair DNA damage. In case the cell cannot repair the damage, p53 is involved in apoptosis initiation. Its pathogenic germline alterations lead to Li-Fraumeni syndrome, and somatic mutations contribute to carcinogenesis in general (Thomas et al, 2022). Analysis of the TCGA-OV dataset confirmed a high *TP53* somatic mutation rate in EOC in general (83% compared with 78% in our dataset). Corroborating our results, previous reports have also documented high rates of *TP53* mutations in EOC tumors, especially of the HGSC subtype (Ahmed et al, 2010; Patch et al, 2015), and their potential to predict chemoresistance (Brachova et al, 2015). However, recent studies did not find an association of *TP53* mutations with EOC recurrence using the WES approach (Li et al, 2019; Zheng et al, 2021). Targeting of the p53 pathway by the WEE1 inhibitor AZD1775 was shown to improve the efficacy of carboplatin in *TP53*-mutated EOC tumors (Leijen et al, 2016), and other options are currently being discussed (Zhang et al, 2022). It is also important to mention that future investigation should be focused on answering the question of whether *TP53* mutations play a role in the efficacy of EOC maintenance therapy with PARPi (McMullen et al, 2020), where platinum resistance usually, but not exclusively, correlates with resistance to PARPi.

Notably, in the sensitive patients, *TP53* mutations were highly significantly co-segregated with alterations in *TTN* and *MUC5B*. This was not true for resistant patients. Because of their generally very high mutation rates in most cancers, *TTN* alone or other FLAGS (Shyr et al, 2014) have been suggested as potential surrogates for TMB (Jia et al, 2019; Oh et al, 2020) which is predictive in terms of responsiveness to immune checkpoint inhibitors (Rizvi et al, 2018). In agreement with the concept of the positive predictive value of TMB or high mutation load in general, we observed significantly longer OS in patients harboring a higher than the median number of small somatic variants or gains. EOC belongs to cancers with rather lower TMB (defined as >20 mutations per Mb), detected in around 10% of cases (Chalmers et al, 2017), but an increased somatic mutation burden was shown to be higher in long-term compared with short-term survivors with the HGSC subtype (Yang et al, 2018). Thus, we may hypothesize that lower proportions of small somatic alterations or CNVs (not fulfilling the above-defined TMB cutoff) may be useful for estimating the prognosis of EOC patients.

On the other hand, patients having mutational signature SBS6 had significantly poorer OS than those in which we did not find this profile. This effect was apparently because of the higher share of this signature among resistant patients. The presence of two other

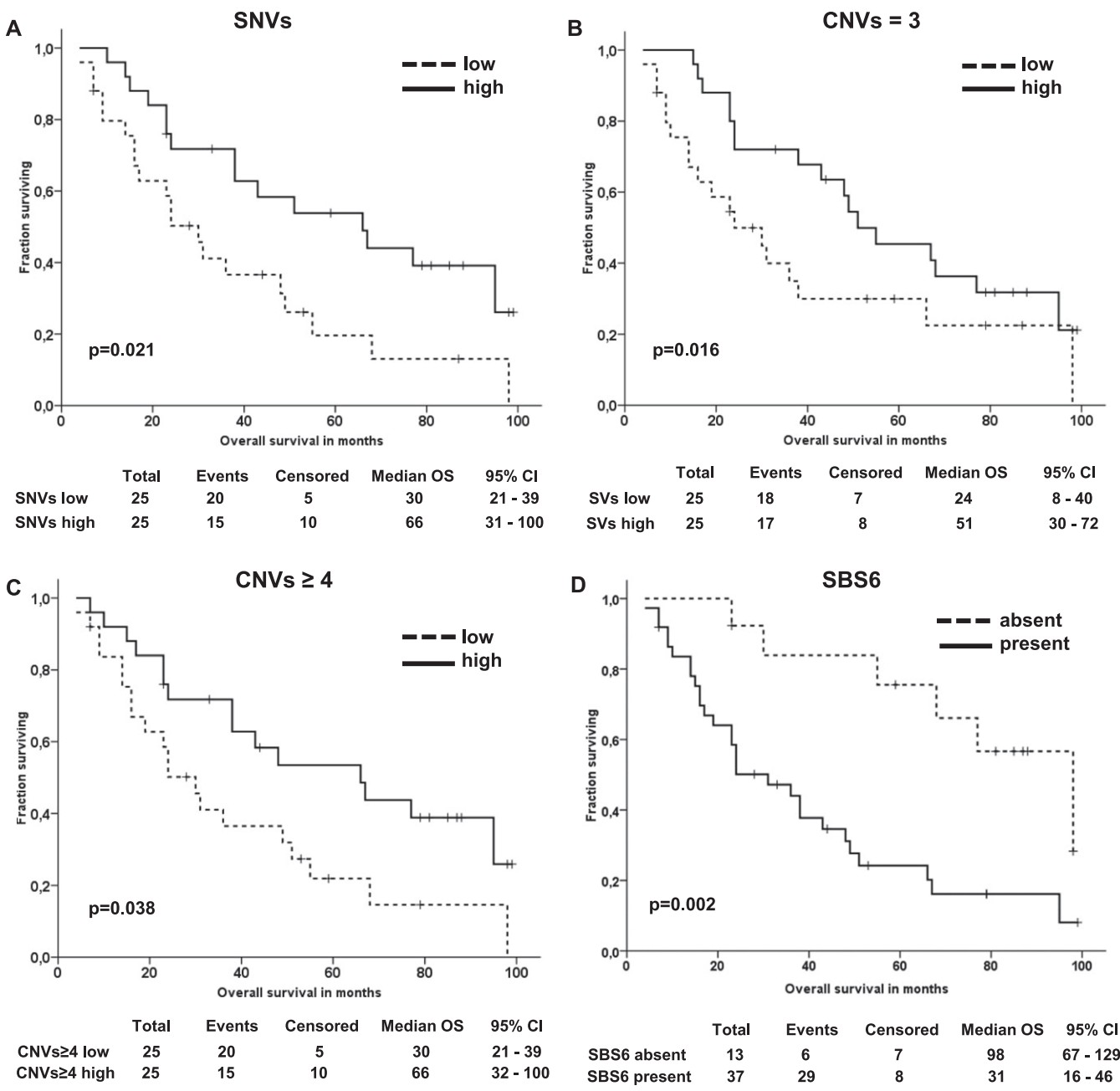

**Figure 6. Kaplan–Meier plots of overall survival of patients.**
**(A, B, C, D)** Patients divided by the median count of somatic variants passing all filters (A), gains with three copies (B), four or more copy number variations (C), and share of SBS6 signature (D).

signatures (SBS10b and SBS21) in patients' tumors showed a trend toward poorer OS as well. This is in contrast to results of Zheng et al (2021) who published significant associations of the SBS10a presence with worse response to platinum and shorter progression-free survival of EOC patients (Zheng et al, 2021). A recently published whole-genome sequencing study reported data on eight selected SBSs representing aging, platinum exposure, and defective homologous recombination (de Witte et al, 2022), of which none associated with the resistance status or survival of EOC patients.

However, most patients analyzed by the previous study were collected at the time of recurrence after heavy systemic pretreatment, which could influence the results, making them not comparable to ours. In addition, we are aware of the considerable constraints of such analyses and the need for further development of robust methods for deciphering and interpreting the biological meaning of mutational signatures (Alexandrov et al, 2020), and thus, these results must be interpreted with extreme caution.

**Table 2. Platinum resistance in patients stratified by alterations in homologous recombination repair genes.**

| Alterations in homologous recombination repair genes | Platinum resistance status | | P-value[a] |
| --- | --- | --- | --- |
| | **Resistant** | **Sensitive** | |
| Somatic mutated | 0 | 7 | 0.033 |
| Somatic non-mutated | 20 | 23 | |
| Germline mutated | 2 | 12 | 0.026 |
| Germline non-mutated | 18 | 18 | |
| Germline or somatic mutated | 2 | 17[b] | 0.001 |
| Germline or somatic non-mutated | 18 | 13[b] | |

[a]P-value by Fisher's exact test.
[b]Two patients had both somatic and germline mutations in any of the homologous recombination repair genes.

On the pathway level, the Hippo pathway was the second most interesting after *TP53*. Mutations in this pathway were significantly less frequent in the resistant patients, and this difference was shared among several genes, for example, *DCHS1*, *PTPN14*, *LLGL1*, *FAT4*, and a few others. The Hippo signaling pathway appears to be highly conserved in eukaryotes and participates in controlling cell proliferation, apoptosis promotion, and stem cell self-renewal. Its kinase module is generally considered tumor suppressive and the transcriptional module tumorigenic (Cunningham & Hansen, 2022). Up-regulation of oncogenic Yes-associated protein (YAP) was connected with multidrug resistance of hepatocellular cancer cell line models in vitro and its targeting, for example, by shRNA restored the chemosensitivity in vivo (Zhou et al, 2019). Another report has shown that the proton pump inhibitor (PPI) esomeprazole, by inhibiting YAP, increased the sensitivity of paclitaxel-resistant EOC subline A2780/T in vitro (He et al, 2019). However, we found a *YAP1*

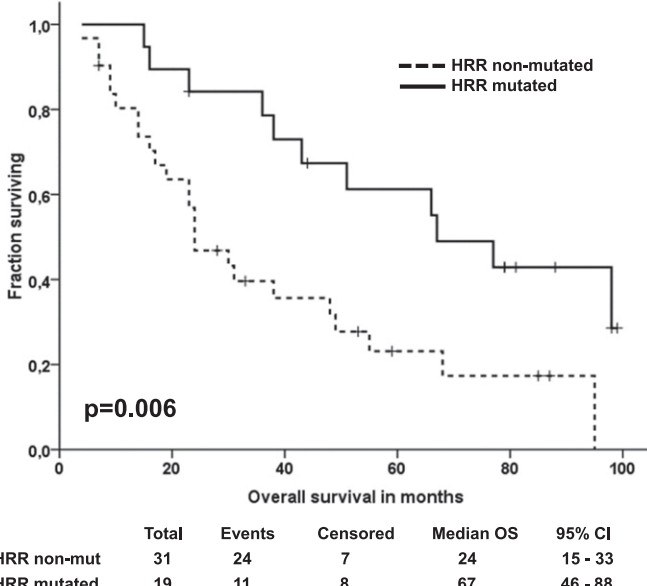

|  | Total | Events | Censored | Median OS | 95% CI |
| --- | --- | --- | --- | --- | --- |
| HRR non-mut | 31 | 24 | 7 | 24 | 15 - 33 |
| HRR mutated | 19 | 11 | 8 | 67 | 46 - 88 |

**Figure 7. Kaplan–Meier plots of overall survival of patients divided by homologous recombination repair alterations.**
Footnote: Patients divided by occurrence of somatic or germline variants in 16 homologous recombination repair genes with functional prediction classified as HIGH.

mutation only in one patient, whereas protocadherin *DCHS1* was mutated in eight sensitive patients versus two resistant patients. In addition, the protein tyrosine phosphatase *PTPN14* was mutated in four sensitive patients but none of the resistant patients. Thus, the data highlight the difficulty of targeting this pathway based on the genetic background alone. For deeper insight, integrative genomic-epigenomic analyses may be necessary.

In the resistant subgroup of patients, mutations in *PABPC3* were less frequent than in the sensitive patients and significantly co-segregated with *PABPC1* alterations. PABPC1 and 3 are members of a highly conserved family of cytoplasmic poly(A)-binding proteins, playing important roles in context-dependent mRNA poly A tail coating (transcript protection) and translation promotion during mammalian development (Xiang & Bartel, 2021). *PABPC3* was recently listed among the most mutated genes in malignant ovarian germ cell tumors (n = 10) (Chen et al, 2021). In addition, PABPC1 transcript up-regulation in EOC tumor samples compared with nonmalignant control tissues and in EOC patients with short OS was found by an in silico analysis of publicly accessible databases, suggesting its potential value as a prognostic biomarker (Feng et al, 2021). *PABPC1* is listed among Tier 2 oncodrivers in the Cancer Gene Census (CGC; https://cancer.sanger.ac.uk/census). Our data suggest that the lack of mutations in *PABPC1* and *PABPC3* may associate with a better response of EOC patients to chemotherapy including platinum derivatives. Other studies reported associations of mutations in different individual genes with platinum resistance and/or EOC prognosis as well. For example, loss of heterozygosity in *FAM175A* was associated with platinum sensitivity and longer progression-free survival of EOC patients (Zheng et al, 2021), and amplifications of *c-MYC* and *PIK3CA* were enriched in EOC recurrent tumors (Li et al, 2019). Whether these associations are causative or correlative remains to be established.

The HGSC tumor subtype represented most of our EOC cohort (70%), and thus, we compared mutational profiles of patients with the HGSC subtype to patients with any other subtype. HGSC tumors had a significantly higher overall mutational load than the rest and more mutations in *TP53*. On the *TP53* pathway level, patients with mutations had almost 10-times higher risk of having the HGSC subtype, an association that passed the FDR adjustment for multiple comparisons. In contrast, patients with the HGSC subtype had lower somatic mutation rates in *PABPC1*, *ZNF717*, *RBMX*, *SLC9B1*, and *TFAM* than the rest of the patients. Aside from *PABPC1*, which is a

Tier 2 oncogene, none of these genes appears in the CGC, and except for *TFAM*, we found no data on their roles in the literature on EOC. The mitochondrial transcription factor A (TFAM), which is essential for mitochondrial biogenesis (Picca & Lezza, 2015), was shown to be down-regulated in the multiresistant SKOV-3-R cell subline in vitro and less expressed also in tumors of the clear cell subtype compared with the HGSC (Gabrielson et al, 2014). A more recent study found that cisplatin-sensitive cell lines of an HGSC origin have a higher mitochondrial density and excess of mitochondrial oxidative stress components, correlating with TFAM (and PGC1α) up-regulation (Kleih et al, 2019). Taken together, these data lead to the hypothesis that functional (and up-regulated) TFAM may be useful as a predictive biomarker for improved therapy response to platinum derivatives of selected patients, for example, without *TFAM* mutations or with TFAM overexpression.

Besides the high *TP53* somatic mutational rate, our study also confirmed the prognostic and predictive value of the mutational profile of HRR genes. Somatic or germline alterations in 16 HRR genes, suggested before as prognostic and potentially predictive for stratification into clinical trials (Norquist et al, 2018), indeed predicted platinum resistance and were associated with a significantly better OS of EOC patients in the present study. The proportion of such patients in our study was 38%, whereas others reported 26% using sequencing of the same genes (Norquist et al, 2018) and 33% by a different method (de Witte et al, 2022). The higher mutational rate observed in our study may be caused by the fact that we sequenced both somatic and germline DNA of all patients which was not the case in the other studies. Most of functional germline alterations were found in BRCA genes, whereas in other HRR genes, such alterations were much less common (*BRCA2*, n = 7 > *BRCA1*, n = 4 >> *ATM*, *BRIP1*, and *RAD51C*, n = 1). This observation may suggest that the list of HRR genes for germline testing may need some more development before its implementation into routine practice.

Our study has some limitations. A modest sample size, especially in subgroup analyses, may be seen as the main limitation, and it is the reason we could not make more detailed comparisons of patients divided by the EOC subtype. We designed the study as a trade-off between high coverage, number of samples, and price, but our prior power analysis suggested that more than 100% difference among the compared groups would be detected with almost 90% power, which we considered satisfactory. The WES design brings further limitations for the analysis of large CNVs, especially for analyses of the noncoding part of the genome, which would be better detected by the whole-genome sequencing technology (de Witte et al, 2022). Nonetheless, the high-coverage WES chosen by us brings deeper insight into potentially functional parts of the genome. In addition, we kept in mind also the limited predictive power of available tools and the relative scarcity of clear links between noncoding areas and cancer reported by large consortia quite recently (Rheinbay et al, 2020). On the other hand, we used only fresh frozen samples of the primary disease before any treatment and recurrence, which makes the study more homogeneous compared with its predecessors.

In conclusion, this study confirms the pivotal role of the HRR profile and somatic mutations in *TP53* in platinum resistance of EOC reported by previous studies and adds new information about the connection between the resistance and the Hippo pathway. Somatic mutations in *PABPC1/3* and *TFAM* may also further contribute to the concept of precision medicine of EOC in the future.

# Materials and Methods

### Patients

Samples of surgically resected, chemotherapy-naive, primary EOC tumors were prospectively collected from 50 EOC patients. Blood samples were taken from these patients to perform matched tumor/normal analysis and eliminate germline variants during bioinformatics evaluation of somatic variants. All patients were diagnosed and treated between 2009 and 2017 in the Motol University Hospital in Prague and the University Hospital in Pilsen.

The clinical data, including age, date of diagnosis, FIGO stage, the histological type and grade of the tumor, the presence of distant metastasis or residuum after surgery, oncological treatment, resistance status, and overall survival (OS) were obtained from medical records. The resistance status was defined according to Friedlander et al, that is, based on the platinum-free interval (PFI) measured as the time from cessation of the platinum-based chemotherapy to disease recurrence or progression (Friedlander et al, 2011). Patients with PFI ≤ 6 mo were considered platinum-resistant, and patients with PFI ≥ 12 mo were platinum-sensitive. Four patients had PFI between 7 and 10 mo and were classified as partially platinum-sensitive (Pejovic et al, 2022). These patients were tentatively included in the resistant group, and statistical analyses were performed with and without them. We further provide only consensual results of these separate analyses. The OS was defined as the time elapsed between surgical resection and death of any cause or patient censoring. The main clinical characteristics of the patients are in Table 1. None of the patients had received neoadjuvant chemotherapy. All patients except one completed first-line chemotherapy with regimens based on a combination of platinum derivatives (carboplatin or cisplatin) and taxane paclitaxel. One patient from the sensitive group received platinum monotherapy.

All patients included in the study read and signed the Informed Consent of the Patient. All procedures performed in this study followed the ethical standards of the Institutional Review Boards of the National Institute of Public Health in Prague (approval reference no. NT14056-3) and the University Hospital in Pilsen (approval reference no. 16-29013A) and agreed with the 1964 Helsinki Declaration and its later amendments or comparable ethical standards. The abovementioned Institutional Review Boards also approved the experimental protocol of the study. This article does not contain any studies with animals.

We performed power calculations considering a comparison between a group of 20 resistant patients and a group of 20 sensitive patients. Mutation frequencies were considered as quantitative variables; thus, the two-sample *t* test was used, and study power was calculated for an expected difference of means between these two groups. According to recent literature, a difference between groups can be up to 140% (Tarazona et al, 2020). Therefore, differences of 20–140% were plotted. A difference of 100% yielded a

power of 87%, whereas a difference of 120% had a power close to 100% (Fig S11).

## DNA isolation, quantification, and quality control

DNA from fresh frozen tumor tissues was isolated with the AllPrep DNA/RNA/Protein Mini Kit (QIAGEN) following the manufacturer's protocol. DNA from peripheral blood lymphocytes was isolated and stored according to the published procedure (Topić & Gluhak, 1991). DNA was eluted into nuclease-free water, aliquoted, and stored at −20°C until further use.

DNA was quantified using Quant-iT PicoGreen dsDNA Reagent and Kits (Invitrogen/Thermo Fisher Scientific) and the plate reader Infinite 200 (Tecan Group Ltd.). DNA purity, such as A260/280 and A260/230 ratios, was checked by the NanoDrop Spectrophotometer 2000 (Thermo Fisher Scientific/Thermo Fisher Scientific).

## Library preparation and WES

The libraries were prepared from tumor and blood DNA using the SureSelect XT Low Input for Illumina Kit with the Human All Exon V7 probe set (both provided by Agilent) according to the manufacturer's protocol. The probe set is designed using the GRCh38/hg38 genome assembly, targets only coding regions from RefSeq, CCDS, GENCODE, and USC Known Genes, and has an end-to-end design size of 48.2 MB. Briefly, 100 ng of tumor or blood DNA were first fragmented using the SureSelect Enzymatic Fragmentation Kit (Agilent) and then pooled for hybridization and capture reactions with the probe set (Agilent). The concentration of DNA libraries before hybridization and of the resulting captured library pools was assessed using the KAPA Library Quantification Kit (Roche). Finally, all libraries were pooled in a non-equimolar fashion (90% tumor and 10% of paired blood DNA libraries) and sequenced on the NovaSeq 6000 platform (Illumina Inc.) using two lanes of the NovaSeq 6000 S4 Reagent Kit with 150-bp paired-end reading (Illumina). The intended average on-target coverage was 250× for tumors and 30× for blood samples with anticipated average duplicate rates of 40% and 10–15%, respectively.

## Raw data preprocessing and variant detection

Adapter sequences were removed from raw FASTQ data using the Agilent Genomics NextGen Toolkit (AGeNT) Trimmer v2.0.2 (www.agilent.com). Reads were aligned to the hg38 human reference genome using Burrows–Wheeler Aligner v0.7.17-r1188 (BWA) with the BWA-MEM algorithm (Li, 2014). Duplicate removal was performed using the AGeNT LocdIt v2.0.2 tool, optimal for SureSelect libraries with Agilent molecular barcodes (Agilent). Only distinct reads were used for downstream analyses.

The following steps were performed using the Genome Analysis Toolkit (GATK) (Broad Institute) according to GATK Best Practices (Van der Auwera et al, 2013). Base recalibration was done by BaseRecalibrator and ApplyBQSR tools v4.1.6.0. To account for Czech population–specific genetic variants, the National Center for Medical Genomics (https://ncmg.cz/en) was used during base recalibration. Germinal variants were detected from blood samples of the EOC patients. Somatic variants were detected from EOC

samples, and blood samples were used as matched normal controls in paired analysis.

Identification of somatic variants and short indels was performed using Mutect2 v4.2.4.1. Variants were filtered using Filter-MutectCalls. Only sites with major alternative allele frequency >0.05 and a minimum number of three reads carrying the alternative allele were considered. Variants with one of the following FILTER statuses (based on the FilterMutectCalls tool) were excluded from further analyses: FAIL, base_qual, contamination, low_allele_frac, map_qual, multiallelic, n_ratio, normal_artifact, orientation, panel_of_normals, position, possible_numt, strand_bias, and weak_evidence. Mutations with the PASS criteria were used. Variants with other criteria such as clustered_events, duplicate, fragment, germline, haplotype, slippage, and strict_strand were also considered as passing filters to avoid type II errors because these variants can in fact be true somatic mutations (Table S2).

Germline single nucleotide polymorphisms and short indels were called using HaplotypeCaller v4.2.4.1, scored using CNNScore Variants v4.2.4.1, and filtered using VariantFiltration v4.2.4.1 tools. Low quality calls were excluded based on the following criteria: sequencing coverage depth (DP) < 10×, variant confidence divided by depth (QD) < 2, phred-scaled probability of strand bias (FS) > 60, strand odd ratio (SOR) > 3, root mean square mapping quality (MQ) < 40, u-based z-approximation from the rank sum test for mapping qualities (MQRankSum) < −12.5, u-based z-approximation from the rank sum test for site position within reads (ReadPosRankSum) < −8.

CNVs were detected with the CNVkit v0.9.8 tool using a matched normal sample as a reference for each tumor sample (Talevich et al, 2016). Significant calls were assessed based on the average read depth $\log_2$ ratio values and B-allele frequencies (BAF) of individual segments, methods implemented in the ExomeDepth tool (Plagnol et al, 2012). Assuming theoretical clonal fraction/sample purity to be 40%, deletion should have $\log_2$ ratio < −0.152 and BAF out of interval 0.4–0.6; duplication should have $\log_2$ ratio > 0.138 and BAF out of interval 0.466–0.533. Besides this, all called segments that contained less than three bins or did not show a statistically significant difference of $\log_2$ ratios compared with reference values ($P$ < 0.05 by the $t$ test) were excluded.

## Variant annotation and interpretation

Found variants were annotated using SnpEff v. 5.0e (Cingolani et al, 2012) and converted to mutation annotation format. Multiallelic positions were split into individual variants, and variants with allelic fraction <5% were excluded. The importance of variants was evaluated based on the SnpEff variant effect prediction that assigned one of the following values to each variant: LOW, MODIFIER, MODERATE, or HIGH. For a complete list of somatic variants passing filters, with additional annotation, see Table S16.

Data used for TMB comparisons were obtained from the TCGA MC3 project (Ellrott et al, 2018). For cancer driver prediction, the TCGA (https://www.cancer.gov/tcga) dataset OV-TP was analyzed by MutSig2CV v3.1, obtained from the Broad Firehose using the firehose_get utility (Broad Institute TCGA Genome Data Analysis Center, 2016) and compared with our dataset analyzed by the MutSigCV v1.3 tool using GenePattern software (Reich et al, 2006; Lawrence et al, 2013).

For comparisons of mutation rates between patient groups and for the creation of somatic variant plots, the Maftools 2.10 R/Bioconductor package (Mayakonda et al, 2018) was used. *P*-values were adjusted by the Benjamini–Hochberg FDR method (Benjamini & Hochberg, 1995), and *P*-values ≤ 0.05 were considered significant.

The pathway analysis was performed using the Maftools 2.10 R/Bioconductor package (Mayakonda et al, 2018). The PANTHER (Protein ANalysis THrough Evolutionary Relationships) classification system was used to classify mutated genes using GO terms (molecular function, biological process, cellular component, and pathways) (Mi et al, 2013). KEGG (Kyoto Encyclopedia of Genes and Genomes) was used for exome-wide pathway analyses (Kanehisa & Goto, 2000).

For mutational signature analysis, we used the R Bioconductor package decompTumor2Sig (Krüger & Piro, 2019). SBS frequency profiles in the context of triplets (mutated base, 5′ adjacent base, and 3′ adjacent base) were constructed for all tumor samples under the Alexandrov signature model without taking into account transcription direction. Then the contribution of observed SBS frequencies profiles to the Cosmic SBS96 v. 3.2 mutational signature dataset (https://cancer.sanger.ac.uk/signatures/sbs/) was estimated. Larger motifs, for example, rearrangement signatures or copy number signatures were not addressed because of the limited power of WES and low expected frequencies of these motifs in the study cohort. For evaluation of associations with clinical data, individual SBSs with >10% of nonzero values across all tumor samples, were used.

### Statistical analysis

Unsupervised exploratory analyses of somatic variants and pathway enrichment in all patients or those divided by resistance status and histological subtype were performed using the package Maftools (Fisher's exact test) (Mayakonda et al, 2018). Differences in mutation load considering whole-exome data or selected gene groups, mutational signatures, and CNVs (both mean size and numbers of patients with different CNV classes) between patients, stratified by clinical data, were compared using the Pearson's chi-square test (for factorial comparisons) or the Mann–Whitney test (factorial versus continuous data). Correlations of continuous data were assessed using the Spearman's $\rho$ test. Survival functions for groups of patients divided by genetic and clinical data were plotted using the Kaplan–Meier method, and significance was calculated by the Breslow test. All continuous variables were divided by the median except SBS signatures which we evaluated as a group with zero values against those with nonzero values. A two-sided *P*-value of less than 0.05 was considered significant. The SPSS v16 program (SPSS Inc.) was used for statistical analyses. Correction to multiple testing was performed by the Benjamini–Hochberg FDR method (Benjamini & Hochberg, 1995).

## Data Access

All data generated or analyzed during this study are included in this published article. Sequencing data aligned to the canonical hg38 reference genome (BAM files) were submitted to Sequence Read Archive (SRA) under the BioProject ID: PRJNA814851.

## Supplementary Information

## Acknowledgements

This work was funded by the Czech Health Research Council grant no. NV19-08-00113 to P Souček and Cooperation program no. 207035, "Maternal and Childhood Care," 3rd Faculty Medicine, Charles University to L Rob. The authors would like to thank all participating patients for their kind consent to the study and clinical personnel for outstanding support. We are also grateful to the Operational Program Research, Development and Education of the National Center for Medical Genomics (reg. No. CZ.02.1.01/0.0/0.0/18_046/0015515) for providing data on the germline profile of the Czech population for comparison.

### Author Contributions

V Hlaváč: conceptualization, investigation, methodology, and writing—original draft, review, and editing.
P Holý: formal analysis, investigation, methodology, and writing—original draft, review, and editing.
R Václavíková: data curation, formal analysis, and writing—review and editing.
L Rob: resources, data curation, and writing—review and editing.
M Hruda: resources, data curation, and writing—review and editing.
M Mrhalová: data curation, formal analysis, and writing—review and editing.
P Černaj: resources, data curation, and writing—review and editing.
J Bouda: resources, data curation, and writing—review and editing.
P Souček: conceptualization, formal analysis, supervision, funding acquisition, and writing—original draft, review, and editing.

### Conflict of Interest Statement

The authors declare no conflict of interest. The funders had no role in the design of the study; in the collection, analyses, or interpretation of data; in the writing of the manuscript; or in the decision to publish the results.

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
