## [Reviewer comments · Life Science Alliance]

Life Science Alliance

Whole Exome Sequencing of Epithelial Ovarian Carcinomas Differing in Resistance to Platinum Therapy

Viktor Hlavac, Petr Holy, Radka Vaclavikova, Lukas Rob, Martin Hruda, Marcela Mrhalova, Petr Cernaj, Jiri Bouda, and Pavel Soucek

DOI: <https://doi.org/10.26508/lsa.202201551>

Corresponding author(s): Pavel Soucek, Charles University

Review Timeline:

Submission Date:	2022-06-07
Editorial Decision:	2022-08-26
Revision Received:	2022-09-09
Editorial Decision:	2022-09-22
Revision Received:	2022-09-23
Accepted:	2022-09-23

Scientific Editor: Novella Guidi

Transaction Report:

August 26, 2022

Re: Life Science Alliance manuscript #LSA-2022-01551-T

Pavel Soucek
Biomedical Center, Faculty of Medicine in Pilsen, Charles University

Dear Dr. Soucek,

Thank you for submitting your manuscript entitled "Whole Exome Sequencing of Ovarian Carcinomas Differing in Resistance to Platinum Derivatives" to Life Science Alliance. The manuscript was assessed by expert reviewers, whose comments are appended to this letter. We invite you to submit a revised manuscript addressing the Reviewer comments.

Thank you for this interesting contribution to Life Science Alliance. We are looking forward to receiving your revised manuscript.

Sincerely,

B. MANUSCRIPT ORGANIZATION AND FORMATTING:

Reviewer #1 (Comments to the Authors (Required)):

I read with an interest a paper by Hlavac and co-investigators studying the whole exome sequencing in DNA of OC patients. A study group of 50 blood and tissue tumor samples were selected. Molecular genetic analysis has been described in details. In my opinion, an interesting note is that "...patients resistant to adjuvant chemotherapy based on platinum and taxane combination have a significantly higher load of somatic mutations in the TP53 gene than sensitive patients...". Moreover, alterations within the Hippo pathway and mutations of PABPC3 gene were also investigated in details. At a final size, the Authors also discussed several study limitations, including sample size. In my opinion, this study is truly interesting and it is worth publishing. The Authors may also prepare the review article concerning their and well as others studies of whole exome sequencing in OC patients. The list of references should be checked carefully. I recommend for the future study to add, if available, more non-HGSC samples, the number of HGSC cases and non-HGSC cases is not similar (35 vs. 14).

Reviewer #2 (Comments to the Authors (Required)):

The aim of this study was to identify germline and somatic genetic differences in patients with platinum-resistant and platinum-sensitive epithelial ovarian cancer (EOC) and in patients with different molecular subtypes of EOC. The results showed that the landscapes of germline and somatic mutations in different subgroups of EOC patients were associated with their prognosis. Unfortunately, I do not think it is appropriate to publish it in its current form. The main reasons are as follows:

1. Ovarian cancer (OC) in the manuscript should be changed to epithelial ovarian cancer (EOC). Also, why is Ovarian Carcinomas used in the title, as opposed to in the manuscript?
2. The writing of the "introduction" lacks a clear thesis statement.
3. Patients with EOC with a PFI of 6 months or more are considered to have platinum-sensitive disease. Patients with a PFI of less than 6 months are considered to have platinum-resistant disease. Are the definitions of resistant and sensitive patients in this manuscript correct?
4. The results showed a significantly higher somatic mutation rate in the TP53 gene in platinum-resistant patients. The role of somatic mutations in the homologous recombination repair (HRR) gene in the platinum sensitivity and favorable prognosis of OC patients was also confirmed. These results make up a large part of the manuscript, but they seem to be well known.
5. It is noted that your manuscript needs to be carefully edited by someone with technical English editing expertise.

Reviewer #3 (Comments to the Authors (Required)):

In this study, Hlavac and Dr. Soucek present the mutational landscape of ovarian cancer. They performed whole exome sequencing in 50 pairs of T/N ovarian cancer patients, 20 out of 50 patients are platinum resistant, and 30 are platinum sensitive. They did a thorough comparison between these two groups: mutation profile, signaling pathway, CNVs and HRR genes. They reported that patients with platinum resistance have a significantly higher load of somatic mutations in TP53 gene. Also, Hippo pathway and HRR genes are with lower mutation burden in resistant patients.

Till now, there is no definite biomarker of platinum resistance in ovarian cancer. Therefore, such research trying to dig out the genetical difference between chemotherapy-resistant and -sensitive groups is with high potential value and interest. The paper is well drafted, but there are some critical missing values in this paper that are necessary on validating the conclusion:

1. The description on filtering the somatic variants is not clear. I cannot find section 2.4. Also, the criteria of filtering reliable variants is kind of not very stringent, ≥ 3 minimum variant reads is not good enough for fresh frozen tissue samples. I also don't know whether the reads are total reads or distinct reads. A supplemental table with all the mutations (with Mutation allele frequency information) passed filter is needed.
2. Similar studies have been performed previously. In 2018, Charles Li did the similar study in ovarian cancer, also reported TP53 mutations, but found no difference between primary and recurrent OC. (PMID: 30584090 PMID: PMC6329978 DOI: 10.1073/pnas.1814027116); Zheng et al. also reported sensitivity predictive model of epithelial ovarian cancer. They reported FAM175A LOH in platinum-sensitive group (PMCID: PMC8481766 PMID: 34604063 doi: 10.3389/fonc.2021.725264). Discussion between the current study and those previous reports is highly recommended.

Ms. No. LSA-2022-01551-T: "Whole Exome Sequencing of Epithelial Ovarian Carcinomas Differing in Resistance to Platinum Derivatives" by Hlavac et al.

Authors' reply to reviewers' comments

We would like to kindly thank the Reviewers for their supportive feedback on the manuscript. We have now revised it to include all the points suggested by the Reviewers and truly hope that the contents are now more clear and fully informative for readers. All changes are highlighted in red letter mode in the revised manuscript and below are our responses to Reviewers' comments.

Reviewer #1 (Comments to the Authors (Required)):

I read with an interest a paper by Hlavac and co-investigators studying the whole exome sequencing in DNA of OC patients. A study group of 50 blood and tissue tumor samples were selected. Molecular genetic analysis has been described in details. In my opinion, an interesting note is that "...patients resistant to adjuvant chemotherapy based on platinum and taxane combination have a significantly higher load of somatic mutations in the TP53 gene than sensitive patients...".

Moreover, alterations within the Hippo pathway and mutations of PABPC3 gene were also investigated in details. At a final size, the Authors also discussed several study limitations, including sample size. In my opinion, this study is truly interesting and it is worth publishing. The Authors may also prepare the review article concerning their and well as others studies of whole exome sequencing in OC patients.

Authors' response: We would like to thank Reviewer 1 for encouraging comments and for the suggestion to create a review article. We agree that such an article would be of general interest, since there are very few general reviews of recent WES results in OC and most reviews focused on a limited number of discussed genes/variants, primarily in the clinical context, not in the basic science context. We plan to prepare such an article in the near future.

The list of references should be checked carefully.

Authors' response: We have checked the references carefully and harmonized them with the LSA citation style.

I recommend for the future study to add, if available, more non-HGSC samples, the number of HGSC cases and non-HGSC cases is not similar (35 vs. 14).

Response: We absolutely agree with this point! Indeed, targeted study on reasonable number of sensitive and resistant non-HGSC, especially LGSC, patients would be of enormous importance due to the scarcity of this subtype, the demonstrated lower response rate to chemotherapy and recently reported activity of targeted therapy with MEK inhibitors. We would like to perform such a study, but we must admit that it is currently more accessible for large multi-centric consortia. We primarily designed the study to compare resistant and sensitive patients, with other comparisons being secondary. At this point, we have around 20 LGSC and wait for classification of resistance of some of them. Once we reach a reasonable study power, we would like to proceed with analyses. The other non-HGSC subtypes are even rarer and represent a serious challenge.

Reviewer #2 (Comments to the Authors (Required)):

The aim of this study was to identify germline and somatic genetic differences in patients with platinum-resistant and platinum-sensitive epithelial ovarian cancer (EOC) and in patients with different molecular subtypes of EOC. The results showed that the landscapes of germline and somatic mutations in different subgroups of EOC patients were associated with their prognosis. Unfortunately, I do not think it is appropriate to publish it in its current form. The main reasons are as follows:

1. Ovarian cancer (OC) in the manuscript should be changed to epithelial ovarian cancer (EOC). Also, why is Ovarian Carcinomas used in the title, as opposed to in the manuscript?

Authors' response: We would like to thank Reviewer 2 for the valuable comment. We revised the title and running title to more accurately reflect the contents of the article. We changed the term ovarian cancer to epithelial ovarian carcinoma (EOC). Noun "carcinoma" in the title and elsewhere reflects merely patient tissues, whereas a more general expression "cancer" is now used only for description of the disease epidemiology in the Introduction and general information.

2. The writing of the "introduction" lacks a clear thesis statement.

Authors' response: We included such a statement at the end of the general introduction and before study aims.

3. Patients with EOC with a PFI of 6 months or more are considered to have platinum-sensitive disease. Patients with a PFI of less than 6 months are considered to have platinum-resistant disease. Are the definitions of resistant and sensitive patients in this manuscript correct?

Authors' response: Reviewer 2 is right. Currently the resistance status is broadly debated. The previously used classification based on 6 months only does not seem to be enough in light of results of recent clinical trials. Nowadays, an even more detailed division is proposed, i.e. PFI up to 3 months = platinum-refractory, 3 to 6 = platinum-resistant, 6 to 12 months = partially platinum-sensitive and the rest sensitive (Pejovic et al. 2022, doi: [10.20517/cdr.2021.138](https://doi.org/10.20517/cdr.2021.138)). Such division atomizes the study set as refractory patients are rare. For the sake of study power we

included four partially platinum-sensitive patients (PFI between 7 and 10 months) into the resistant subgroup. We performed statistical analyses with and without them and further provide only consensual results of these separate analyses. We had reformulated the definition to make the description clearer and added the most recent reference (Section Patients in Chapter Materials and Methods).

4. The results showed a significantly higher somatic mutation rate in the TP53 gene in platinum-resistant patients. The role of somatic mutations in the homologous recombination repair (HRR) gene in the platinum sensitivity and favorable prognosis of OC patients was also confirmed. These results make up a large part of the manuscript, but they seem to be well known.

Authors' response: We partly agree with Reviewer 2 that some results are confirmatory, but we consider such agreement with already published results as indication that the study was well conducted. On the other hand, we found a number of original never published results. For example: PABPC1/3 and TFAM individual gene mutations were never published in the context of EOC as well as enrichment of Hippo pathway mutations in sensitive patients. These patients, despite being sensitive and a good prognosis subgroup often experience relapse. There are over 30 patented modulators of the Hippo pathway with different mechanisms of action (Zagiel et al. 2022, doi: 10.1080/13543776.2022.2096436.), which may be further tested in this subgroup of tumors (models) and hopefully later also in the patients. We admit that the prognostic role of SBS6 suggested by us needs proper validation, but pilot data are promising ($p=0.002$) and the reverted trend opposed to overall mutational load and CNVs seems intriguing to us. Even the well-known TP53 role in EOC has not been so far thoroughly reflected in targeted therapy and we still need more information about the nature of mutations (GoF vs LoF). Therefore, we feel that further studies contributing to the mosaic of genetic background of platinum resistance are important.

5. It is noted that your manuscript needs to be carefully edited by someone with technical English editing expertise.

Authors' response: We thank the Reviewer 2 for pointing to some style and grammar problems and apologize for inconvenience. The revised paper was carefully edited to improve readability.

Reviewer #3 (Comments to the Authors (Required)):

In this study, Hlavac and Dr. Soucek present the mutational landscape of ovarian cancer. They performed whole exome sequencing in 50 pairs of T/N ovarian cancer patients, 20 out of 50 patients are platinum resistant, and 30 are platinum sensitive. They did a thorough comparison between these two groups: mutation profile, signaling pathway, CNVs and HRR genes. They reported that patients with platinum resistance have a significantly higher load of somatic mutations in TP53 gene. Also, Hippo pathway and HRR genes are with lower mutation burden in resistant patients.

Till now, there is no definite biomarker of platinum resistance in ovarian cancer. Therefore, such research trying to dig out the genetical difference between chemotherapy-resistant and -sensitive groups is with high potential value and interest. The paper is well drafted, but there are some critical missing values in this paper that are necessary on validating the conclusion:

Authors' response: We would like to thank Reviewer 3 for positive feedback.

1. The description on filtering the somatic variants is not clear. I cannot find section 2.4.

Authors' response: We apologize for the error which was a leftover from a previous draft. The filtering of somatic variants is described in the section "Methods", chapter "Raw data preprocessing and variant detection". We have made the description clearer and fixed the reference in Results accordingly.

Also, the criteria of filtering reliable variants is kind of not very stringent, ≥ 3 minimum variant reads is not good enough for fresh frozen tissue samples.

Authors' response: While for clinical diagnostics, this criterion has to be much more stringent, there is no established optimal setting for experimental studies. For our purpose, where we do not evaluate specific individual variants, but instead general trends, mutation burden, signatures, etc., this setting instead allows to shift the balance slightly towards discovering more true positives at the cost of introducing a negligible amount of false positives. Despite this, the counts of supporting distinct reads of alternate allele across all variants/samples were min. 3 - max. 711 and 45.7 on average (median 33). Most importantly, we followed variant calling with a major main filtration step utilizing the FilterMutectCalls function, allowing for removal of variants that are likely false positive, based on multiple GATK statistics (base/mapping quality, presence in a panel of normals, allele fraction clustering model, strand artifact model, etc.). We further clarified this point in the Method chapter, section Raw data preprocessing and variant detection. Last but not least, we believe that with lower stringency of a large number of reads and deep coverage available, we could touch intra-tumoral heterogeneity, which would be missed by a highly stringent approach and hardly can be followed by looking into data in databases such as TCGA. This adds value to our data.

I also don't know whether the reads are total reads or distinct reads.

Authors' response: The filtering step was performed after the removal of duplicates; therefore, the reads were distinct reads. We have provided this information in the Methods and Results sections of the manuscript and apologize for omitting this important information in the previous version.

A supplemental table with all the mutations (with Mutation allele frequency information) passed filter is needed.

Authors' response: A Supplemental Table S16 with all mutations that passed filter including tumor allele frequency has been added to the revised submission. A reference to the table has been added to the Methods section.

2. Similar studies have been performed previously. In 2018, Charles Li did the similar study in ovarian cancer, also reported TP53 mutations, but found no difference between primary and recurrent OC. (PMID: 30584090 PMID: PMC6329978 DOI: 10.1073/pnas.1814027116); Zheng et al. also reported sensitivity predictive model of epithelial ovarian cancer. They reported FAM175A LOH in platinum-sensitive group (PMCID: PMC8481766 PMID: 34604063 doi: 10.3389/fonc.2021.725264). Discussion between the current study and those previous reports is highly recommended.

Authors' response: We are grateful to the Reviewer 3 for tips on improving the Discussion section.

Additional note: During the revision process, we noticed that one figure (Fig. 5) describing the difference between subtypes was incorrect in the manuscript and the information about the PABPC1 mutation frequency in HGSC subtype was reverted. We corrected this both in the Results section (Fig. 5 and sentence below) and Discussion about subtypes. We deeply apologize to the Reviewers for this oversight!

September 22, 2022

RE: Life Science Alliance Manuscript #LSA-2022-01551-TR

Prof. Pavel Soucek
Charles University
Faculty of Medicine in Pilsen, Biomedical Center
Laboratory of Pharmacogenomics
alej Svobody 1655/76
Pilsen 323 00
Czech Republic

Dear Dr. Soucek,

Thank you for submitting your revised manuscript entitled "Whole Exome Sequencing of Epithelial Ovarian Carcinomas Differing in Resistance to Platinum Therapy". We would be happy to publish your paper in Life Science Alliance pending final revisions necessary to meet our formatting guidelines.

- please upload your supplementary figures as single files
- please add ORCID ID for corresponding author-you should have received instructions on how to do so
- please double-check your figure legends for Figure S6, Figure S10, and Figure S11 and add the panels A and B to the legend as well as to the figure callouts in the main manuscript text

A. FINAL FILES:

B. MANUSCRIPT ORGANIZATION AND FORMATTING:

Sincerely,

Reviewer #4 (Comments to the Authors (Required)):

The authors made a dramatical improvement on the raw data part and also the discussion part. They answered most of my questions. I recommend this paper for publication.

September 23, 2022

RE: Life Science Alliance Manuscript #LSA-2022-01551-TRR

Prof. Pavel Soucek
Charles University
Faculty of Medicine in Pilsen, Biomedical Center
Laboratory of Pharmacogenomics
alej Svobody 1655/76
Pilsen 323 00
Czech Republic

Dear Dr. Soucek,

Thank you for submitting your Research Article entitled "Whole Exome Sequencing of Epithelial Ovarian Carcinomas Differing in Resistance to Platinum Therapy". It is a pleasure to let you know that your manuscript is now accepted for publication in Life Science Alliance. Congratulations on this interesting work.

DISTRIBUTION OF MATERIALS:

Again, congratulations on a very nice paper. I hope you found the review process to be constructive and are pleased with how the manuscript was handled editorially. We look forward to future exciting submissions from your lab.

Sincerely,
